# Combinatorial Gene Expression Profiling of Serum *HULC*, *HOTAIR*, and *UCA1* lncRNAs to Differentiate Hepatocellular Carcinoma from Liver Diseases: A Systematic Review and Meta-Analysis

**DOI:** 10.3390/ijms25021258

**Published:** 2024-01-19

**Authors:** Lalita Lumkul, Phatcharida Jantaree, Kritsada Jaisamak, Wasinee Wongkummool, Worakitti Lapisatepun, Santhasiri Orrapin, Sasimol Udomruk, Luca Lo Piccolo, Parunya Chaiyawat

**Affiliations:** 1Center of Multidisciplinary Technology for Advanced Medicine, Faculty of Medicine, Chiang Mai University, Chiang Mai 50200, Thailand; lalita.l@cmu.ac.th (L.L.); phatcharida.j@cmu.ac.th (P.J.); kritsada.jaisamak@gmail.com (K.J.); wasinee.won@cmu.ac.th (W.W.); santhasiri.or@cmu.ac.th (S.O.); sasimol.ud@cmu.ac.th (S.U.); 2Center for Clinical Epidemiology and Clinical Statistics, Faculty of Medicine, Chiang Mai University, Chiang Mai 50200, Thailand; 3Department of Surgery, Faculty of Medicine, Chiang Mai University, Chiang Mai 50200, Thailand; worakitti.l@cmu.ac.th

**Keywords:** hepatocellular carcinoma, liquid biopsy, diagnostic biomarkers, long non-coding RNA

## Abstract

Hepatocellular carcinoma (HCC) presents a significant global health challenge due to limited early detection methods, primarily relying on conventional approaches like imaging and alpha-fetoprotein (AFP). Although non-coding RNAs (ncRNAs) show promise as potential biomarkers in HCC, their true utility remains uncertain. We conducted a comprehensive review of 76 articles, analyzing 88 circulating lncRNAs in 6426 HCC patients. However, the lack of a standardized workflow protocol has hampered holistic comparisons across the literature. Consequently, we herein confined our meta-analysis to only a subset of these lncRNAs. The combined analysis of serum *highly upregulated in liver cancer* (HULC) gene expression with *homeobox transcript antisense intergenic RNA* (HOTAIR) and *urothelial carcinoma-associated 1* (UCA1) demonstrated markedly enhanced sensitivity and specificity in diagnostic capability compared to traditional biomarkers or other ncRNAs. These findings could have substantial implications for the early diagnosis and tailored treatment of HCC.

## 1. Introduction

Hepatocellular carcinoma (HCC) is the most common liver cancer subtype, accounting for 75–85% of all liver malignancies. Globally, it ranks as the third leading cause of cancer-related mortality, with approximately 900,000 new cases and 830,000 reported deaths [1]. The primary causes of HCC are chronic infection with hepatitis B virus (HBV) or hepatitis C virus (HCV). Furthermore, the progression of this disease can be accelerated by various risk factors, including cirrhosis, heavy alcohol consumption, aflatoxin-contaminated foods, non-alcoholic fatty liver disease (NAFLD), and smoking [1,2]. While preventive measures against viral infections have proven effective in reducing HCC incidence in certain regions, high-income countries are experiencing an uptick in cancer prevalence due to factors such as population aging, increased body weight, and diabetes [3,4].

The early detection and timely management of HCC are crucial to reduce the economic burden of treatment and improve patient outcomes [3]. Surveillance initiatives directed at high-risk groups have become increasingly important for this purpose [5]. Although surveillance programs offer notable advantages, they require thorough assessment, weighing their benefits against potential downsides. Non-invasive screening tools such as ultrasound (US) and serum biomarker alpha-fetoprotein (AFP) have been incorporated into these programs [6]. However, their efficacy in detecting early-stage HCC is limited, particularly with US exhibiting only 63% sensitivity, and even more challenging in patients with chronic liver disease, i.e., cirrhosis [7]. Combining US with AFP is proposed using several screening criteria; however, there is difficulty in AFP result interpretation itself, which is confounded by hepatitis viral infection, in addition to the cost increment and higher number of false positive cases [7,8,9]. Other than this, people with false positive results would experience surveillance-related physical harm in diagnosis procedures, such as computed tomography (CT), magnetic resonance imaging (MRI), and biopsies [10]. These methods have been associated with adverse effects, including physical, financial, and psychological burdens, on patients, as well as invasive procedures leading to potential complications [10].

Consequently, non-invasive biopsy has become an attractive choice for the early detection of cancer. Various biomarkers for liquid biopsy in HCC have been explored, including cell-free nucleic acids (cfNAs), extracellular vesicles, and circulating tumor cells [11]. The stability, sensitivity, and ease of detection and quantification make cfNAs, such as circulating tumor DNA (ctDNA) and non-coding RNAs (e.g., miRNAs and lncRNAs), advantageous for certain applications like early cancer detection and monitoring [12].

LncRNAs are a group of RNA molecules comprising at least 200 nucleotides (nts) and possessing little to no coding potential. Similar to protein coding RNAs (mRNAs), lncRNAs are transcribed independently via RNA Polymerase II (Pol II), often undergoing capping, splicing, and polyadenylation. While certain lncRNAs are expressed at lower levels than mRNAs and exhibit tissue- and cell-specific expression patterns, others are abundant and found in various cell types [13,14]. Remarkably, lncRNAs exhibit both structural and regulatory features, even though the precise function of cell-free circulating lncRNAs remains unclear [15,16].

Several circulating long non-coding RNAs (lncRNAs) have shown promise as novel diagnostic markers for hepatocellular carcinoma (HCC) [17,18,19,20], but the lack of standardization, validation, biological understanding, and regulatory approval pose challenges to widespread clinical use. In this study, we conducted a systematic search and comprehensive analysis of the diagnostic performance of circulating lncRNAs in HCC compared to healthy controls (HCs) or other liver diseases (LDs). To address major limitations, we specifically identified lncRNAs that met certain criteria: their expression was measured in the same type of body fluid, compared using the same housekeeping gene, and accompanied by sensitivity and specificity data to distinguish HCC from LDs. Subsequently, we performed a meta-analysis to extract pertinent diagnostic features of these identified circulating lncRNAs in HCC.

## 2. Materials and Methods

This systematic review was conducted following the recommendation of the *Cochrane Collaboration Handbook* for systematic reviews of diagnostic test accuracy [21]. Our prespecified protocol has been registered at the International Prospective Register of Systematic Reviews (PROSPERO: CRD42022363196), and the ethical exemption was approved due to secondary data utilization by the Ethical Committee of the Faculty of Medicine, Chiang Mai University (EXEMPTION 9255/2022, FAC-MED-2565-09255). The study was reported in line with the Preferred Reporting Items for Systematic Review and Meta-Analysis (PRISMA) statement (Appendix A).

### 2.1. Systematic Searching and Eligible Criteria

We conducted a comprehensive search through electronic medical databases including PubMed, EMBASE, and Scopus from their inception until 7 December 2022 with no language restriction. Search strategies and obtaining records are described in Appendix A. Deduplication of records was performed in citation manager, and the screening process of relevant titles and abstracts was conducted in Rayyan by two independent authors (L.L. and L.L.P.). The remaining records were retrieved, and full-text articles were evaluated for eligibility (L.L. and K.J.). The inclusion criteria were case-control or cohort studies presenting comparative data of human biospecimens obtained from liquid biopsy, including peripheral blood, plasma, or serum of HCC and control subjects. We excluded records that (i) were case series/case reports, reviews, commentaries, letters to editor and (ii) investigated the expression in cell compartments (i.e., peripheral blood mononucleated cells). Any discrepancies in screening processes were resolved by discussion with the third reviewer. Reasons for exclusion of each full-text record are listed in Appendix A.

### 2.2. Outcome of Interest

The expression level of circulating lncRNAs and their diagnostic performance in discriminating HCC from liver disease (LD) patients were retrieved from a systematic literature search. The HCC patients, with or without viral infection, could be diagnosed based on imaging of the liver using computed tomography (CT) and/or dynamic magnetic resonance imaging. The stage of HCC could be classified according to the Barcelona Clinic Liver Cancer (BCLC) classification. LD group was defined as high-risk patients in whom surveillance is recommended based on the EASL Clinical Practice Guidelines: Management of hepatocellular carcinoma [7]. This group includes patients with liver disease with cirrhosis and without cirrhosis but with hepatitis viral infection.

### 2.3. Data Extraction

Data were independently extracted by two reviewers (L.L. and K.J.). For all eligible studies, we extracted study characteristics (i.e., authors, year of publication, study site, candidate lncRNAs), sample preparation (i.e., type of sample, expression level measurement method), sample size, and participant definitions (definition of case and control groups). We counted the frequency of lncRNAs being investigated, and those investigated in more than three articles were further examined in detail. We extracted detailed information from those studies, and the number of participant groups, lncRNA expression levels, housekeeping genes, diagnostic performance including area under receiver operative characteristic curves (auROCs), as well as their cut-off value, sensitivity, and specificity, were collected. Data presented in graphs or bar charts were extracted using WebPlotDigitizer [22].

### 2.4. Quality Assessment

Two reviewers (L.L. and P.J.) determined the quality of studies investigated on candidate lncRNAs using QUADAS-2: A Revised Tool for the Quality Assessment of Diagnostic Accuracy Studies [23]. Risk of bias was evaluated across 4 domains, which were patient selection, index test, reference standard, and flow and timing as high risk, low risk, or unclear. In addition, applicability concerns were similarly assessed across the first 3 domains. Disagreements were resolved by the third reviewer (L.L.P.) to reach a consensus.

### 2.5. Statistical Analysis

All statistical analyses were conducted using Stata version 16 (StataCorp, College Station, TX, USA) and Microsoft Excel v.16.75.2 (Microsoft Corporation, Redmond, WA, USA). Mean expression levels of candidate lncRNAs were calculated and compared between HCC patients and healthy groups. The effect size was reported using Cohen’s d standardized mean difference (SMD) and 95% confidence interval (CI). Meta-analysis of mean expression level was performed using a random effects model comparing continuous variables. Heterogeneity was estimated based on I-square (I^2^) and Q-statistics. The expression differences based on housekeeping genes were visualized, using a forest plot, as downregulated or upregulated. We calculated true positive, false negative, true negative, and false positive using their sample size, sensitivity, and specificity. Overall diagnostic indices (e.g., summary receiver operating characteristic (SROC) curve, pooled sensitivity and specificity, diagnostic odds ratio) were estimated using midas command in Stata. In the case that more than 10 studies were included in the meta-analysis, sensitivity analysis, subgroup analysis, and publication bias assessment using funnel plot and Egger’s test were conducted as appropriate.

## 3. Results

This study identified 4108 records from three databases. After screening for relevant abstracts and titles, 111 articles were obtained. Of these, we evaluated 107 retrievable full-text studies, and 76 of them were eligible for our data synthesis (Figure 1). All exclusions are listed in Appendix A.

### 3.1. Characteristics of Eligible Studies

The study characteristics of all inclusions are listed in Appendix A. Among 76 studies, 64 (79.0%) studies were conducted in East Asian countries (China and the Republic of Korea), 15 (18.5%) studies were conducted in Egypt, and 2 (2.5%) were conducted in Italy. More than half of them (47 studies, 58.0%) utilized serum samples, and 19 studies (23.5%) used plasma. Exosomes were investigated in both serum and plasma, accounting for seven (8.6%) and four (4.9%) studies, respectively. Whole blood samples were examined in two (2.5%) studies, as well as saliva (2.5%).

All studies were case-control designs investigating the diagnostic potential of lncRNAs in HCC. There was a total of 13,621 participants including 6479 cases and 7142 controls. While cases were HCC (either with or without hepatitis viral infection), controls were recruited as healthy volunteers, LD patients, or cirrhosis patients. There were 34 studies (42%) comparing the expression of lncRNAs in HCC with liver diseases, whereas the others (47 studies) compared the results with only healthy controls.

Only a few serum lncRNAs have been consistently investigated for their association with HCC.

From all studies included, 88 lncRNAs were identified (Appendix A), of which only 6 lncRNAs have been investigated in more than two independent studies or validations, which included *HOTAIR*, *HULC*, *TUG1*, *MALAT1*, *MEG3*, and *UCA1* (hereafter referred to as candidate lncRNAs). Data were extracted from 15 eligible studies on the candidate lncRNAs and are presented in Table 1 and Appendix A. These studies included 937 HCC patients and 387 LDs. The expression levels and diagnostic performances of these lncRNAs were examined using quantitative reverse transcriptase polymerase chain reaction (qRT-PCR) and normalized in comparison to different types of housekeeping genes including 5S or 18S rRNAs, β-actin, Glyceraldehyde-3-Phosphate Dehydrogenase (*GAPDH*), and Hydroxymethylbilane Synthase (*HMBS*).

### 3.2. Quality of Evidence among Candidate lncRNAs

The quality of eligible studies was determined based on QUADAS-2 across four risks of bias domains and three applicability concerns (Figure 2). All studies adopted a case-control design, which is generally necessary for primary investigations into lncRNA expression levels. However, the interpretation of the results raised some concerns. Since the diagnostic study required measurements in all consecutive patients suspected of having the disease to prevent potential bias, the use of a case-control design may result in an overestimation of the outcomes [23].

Additionally, concerning the index test, none of the studies pre-specified the appropriate cut-off (threshold) point for diagnostic accuracy analyses. Consequently, the results exhibited high heterogeneity and were considered data-driven analyses by the nature of the study.

Furthermore, a few studies (13.3%) did not provide adequate diagnostic criteria for patients with HCC and liver disease. Most healthy subjects did not undergo similar reference tests, primarily because these tests are invasive, such as liver biopsy. Additionally, one study (Kim SS et al., 2021) excluded certain patients from the final analysis. Notably, seven studies (46.7%) (Xie et al., 2014; Li et al., 2015; Kamel et al., 2016; Dong et al., 2019; Huang et al., 2020; Shaker et al., 2020; Kim et al., 2021) did not clearly specify whether the HCC samples were collected before any treatment.

In terms of applicability, more than 80% of the studies provided evidence that matches our research question across three applicability domains (Appendix A).

### 3.3. Expression Level of Candidate lncRNAs in HCC Compared to Healthy Controls

Comparing HCC and healthy controls, circulating *HOTAIR* and *HULC* were consistently upregulated regardless of the housekeeping genes used for their normalization (Figure 3). Conversely, conflicting results were obtained for circulating *MALAT1*, *MEG3*, *TUG1*, and *UCA1*, the expression of which in serum has been found to be either up- or downregulated in HCC versus healthy controls, depending on the housekeeping gene used for their normalization (Figure 3). Of these studies, no significant publication bias was observed when Egger’s test of small study effects was performed (Figure 4; *p*-value, 0.208). Nonetheless, we performed a comparative analysis of circulating *UCA1* expression levels by consolidating data from four independent studies that had normalized serum *UCA1* levels to *GAPDH*. In addition, we calculated Cohen’s d effect size, resulting in an estimated overall expression difference of 1.40 (95% CI, 0.70–2.09) between HCC and healthy controls (Figure 5). Altogether, *HOTAIR*, *HULC*, and *UCA1* lncRNAs exhibited upregulation in HCC serum compared to healthy controls when normalized against *GAPDH*.

### 3.4. Diagnostic Performance to Discriminate HCC from Liver Diseases

While a screening test is used to detect conditions in healthy individuals, a diagnostic test is designed to confirm or rule out the health condition of patients at high risk or having signs of symptoms. Thus, we further investigated the discriminative ability of lncRNAs in HCC compared to LD patients, in whom surveillance is recommended. We extracted the diagnostic performance of serum *HOTAIR*, *HULC*, and *UCA1* that contained the most similar measurement condition in HCC compared to patients having LDs. Consequently, the results determined under *GAPDH* normalization were extracted (Table 2). For these four studies, 271 LD patients from Chinese and Egyptian backgrounds were all defined as having chronic HBV infection, cirrhosis, or fatty liver disease. The performance for each study and the pooled performance of serum *HOTAIR*, *HULC*, and *UCA1* are shown in Figure 5. Notably, serum *HULC* [30] demonstrated the best performance, while serum *HOTAIR* and *UCA1* showed comparable results. When considering the overall estimates for serum *HULC* [30] *HOTAIR* [25,26], and *UCA1* [30,38], they collectively exhibit excellent discrimination capabilities between HCC and LD patients (with an area under the SROC curve of 86%; 95% CI, 83–89%). Their sensitivity is approximately 75% (95% CI, 67–82%), and their specificity is 87% (95% CI, 83–89%). The diagnostic odds ratio (DOR) stands at 20 (95% CI, 10–42), albeit with a notable heterogeneity of 81% (I2 81%; 95% CI, 60–100) (Figure 6). These data suggest that a combinatorial gene expression profiling of serum *HULC*, *UCA1*, and *HOTAIR* lncRNAs could provide excellent diagnostic performance to differentiate hepatocellular carcinoma from LD.

## 4. Discussion

The early diagnosis of hepatocellular carcinoma (HCC) has been a subject of intense investigation, with recent studies emphasizing the effectiveness of combining multiple biomarkers. These studies have explored the combination of alpha-fetoprotein (AFP), alongside other biomarkers such as Des-γ-carboxy-prothrombin (DCP), AFP-L3 isoform, serum alanine aminotransferase, serum alkaline phosphatase measurements, and relevant clinical factors like age and sex. These elements are used to construct statistical models, leading to the development of various diagnostic tools, including the GALAD score [39,40], Doylestown algorithm [41], and HES algorithm [42]. More recently, alternative strategies have been considered, involving the use of circulating nucleic acid biomarkers such as miRNAs. However, a valid tool for the early diagnosis of HCC is still lacking.

Our results demonstrated the diagnostic performance of a few circulating lncRNAs as potential non-invasive biomarkers. Notably, *HULC*, *HOTAIR*, and *UCA1* showed promising diagnostic sensitivity and specificity, potentially impacting early HCC diagnosis and treatment. Notably, the AUC of the SROC curve from the pooled studies presented here is higher than that of circulating *miRNAs 141* and the *200a* family, which have been recently proposed as novel liquid biopsy diagnostic markers to distinguish HCC from LDs (with AUC values of 0.75 and 0.73, respectively) [43]. Given these advancements, it is of particular interest to assess the performance of a panel of dysregulated serum long non-coding RNAs (lncRNAs) including serum *HULC*, *HOTAIR*, and *UCA1* in conjunction with statistical models like the GALAD model.

Cell-free nucleic acids are fragments of DNA or RNA that are likely released into the bloodstream by tumor cells [17]. Among these, long non-coding RNAs (lncRNAs) represent the most abundant group and are remarkably resilient to degradation caused by repetitive freeze–thaw cycles, prolonged exposure to 45 °C, and even room-temperature conditions [44,45]. While the precise role of circulating lncRNAs remains unclear, they are widely recognized as crucial regulators of gene expression.

For example, *HULC*, which is specifically associated with HCC, is known to enhance the expression of the *HMGA2* oncogene, stabilize the COX-2 protein, and upregulate sphingosine kinase 1 (SPHK1) [46,47,48,49]. On the other hand, *HOTAIR* and *UCA1* play significant roles in cancer development and progression, primarily through epigenetic mechanisms like miRNA sponging or by recruiting specific chromatin remodelers [50,51].

Altogether, *HULC*, *HOTAIR*, and *UCA1* are implicated in diverse processes linked to carcinogenesis, impacting cell mobility, proliferation, apoptosis, invasion, aggression, and metastasis. However, the physiological or disease mechanisms of their circulating forms remain unclear. Nevertheless, they remain subjects of significant interest in cancer research and hold potential as targets for therapeutic interventions aimed at impeding cancer development and improving patient outcomes.

### 4.1. Study Limitations

Our study presents an updated panel of circulating lncRNAs that hold promise for further implementation in clinical practice for the early diagnosis of HCC. However, it is important to acknowledge a few limitations. Initially, only a limited number of lncRNA types were included from the pool of 88 identified lncRNAs. This decision was influenced by the scarcity of comprehensive information available for an in-depth qualitative synthesis. As a result, the capacity to perform subgroup analyses considering various clinical characteristics, such as age, gender, tumor stage, and lymphatic metastasis, was restricted. These factors could potentially contribute to between-study variations. Therefore, to establish the comprehensive diagnostic utility of lncRNAs in HCC, it is imperative to await further relevant studies and engage in more extensive data analysis.

Furthermore, the risk of bias assessment for diagnostic studies (QUADAS-2) was originally devised with a focus on clinical diagnostic investigations, prioritizing cross-sectional populations and rigorous index and reference tests. Nonetheless, it is worth noting that all the molecular studies included in our analysis utilized a case-control design to examine differential expression levels and conduct additional performance analyses. Although the application of this tool was not entirely straightforward and aligned with the nature of molecular studies, it is recommended that future research endeavors consider aligning their study designs with the recommended procedures for diagnostic studies.

Additionally, the summary receiver operating characteristic (SROC) analysis was conducted individually for various lncRNAs, each with its distinct cut-off point, elucidating the diagnostic performance of each lncRNA as well as the collective performance of the three selected lncRNAs. Consequently, drawing comprehensive overarching conclusions from this array of diverse measurements is not practically feasible. Nevertheless, these outcomes do indicate an encouraging trend toward the utilization of lncRNAs as potential diagnostic biomarkers.

Fourthly, it is noteworthy that a majority of the studies were carried out within Asian populations, while only a limited number of European patients were investigated. Although this may introduce complexities in terms of data generalization, it is important to highlight that our study offers valuable insights into population-specific patterns of lncRNA expression.

### 4.2. Implications and Future Research

Despite the limitations and challenges inherent in the current research, our findings suggest that specific serum lncRNAs, notably *HULC*, *HOTAIR*, and *UCA1*, exhibit promising potential as diagnostic biomarkers for HCC. These lncRNAs display abnormal upregulation in HCC patients, indicating their capacity to distinguish HCC from other conditions.

Future research endeavors in this field should concentrate on several critical areas. First and foremost, validation in larger and more diverse patient cohorts is imperative to confirm the diagnostic potential of these lncRNAs across various populations. This validation process will enhance the reliability and applicability of these biomarkers.

Furthermore, the development of a standardized workflow protocol for lncRNA analysis is essential. This protocol will facilitate comprehensive comparisons across studies, ensuring consistency and reliability in future research endeavors. For instance, adopting a cross-sectional study design that targets individuals scheduled for diagnosis could yield valuable insights. This approach should focus on a population at high risk of HCC, including those with abnormal AFP levels, hepatitis virus infections, chronic liver disease, or presenting symptoms suggestive of undiagnosed HCC. Conducting rigorous exploratory analyses will pave the way for robust findings. To bolster the strength of these findings, it is essential to validate the results in an external population, establishing predefined cut-off points for diagnosis.

While a diagnostic marker’s specificity, sensitivity, and clinical validation are pivotal factors, comprehending the biological roles of circulating lncRNAs can offer valuable context and insights. This biological understanding can assist researchers in selecting and validating the most suitable lncRNAs for diagnostic purposes, thus enhancing the overall effectiveness of clinical diagnostic tools. Currently, some circulating long RNAs have shown stability in blood and diagnostic potential in cancer management. However, we still lack a clear understanding of how these circulating RNAs maintain stability in RNase-rich blood or their specific functions in body fluids. To bridge these knowledge gaps, further studies are essential to uncover their functional roles.

## 5. Conclusions

To date, long non-coding RNAs (lncRNAs) have garnered growing attention as potential diagnostic markers for hepatocellular carcinoma (HCC). Through our meta-analysis, we have observed elevated serum levels of *HULC*, *HOTAIR*, and *UCA1* in HCC patients, indicating that these lncRNAs, when combined with other non-invasive biomarkers, could constitute an enhanced tool for early HCC diagnosis. However, further validation in larger patient cohorts is necessary to assess their potential utility as novel biomarkers in clinical practice.

## Figures and Tables

**Figure 1 ijms-25-01258-f001:**
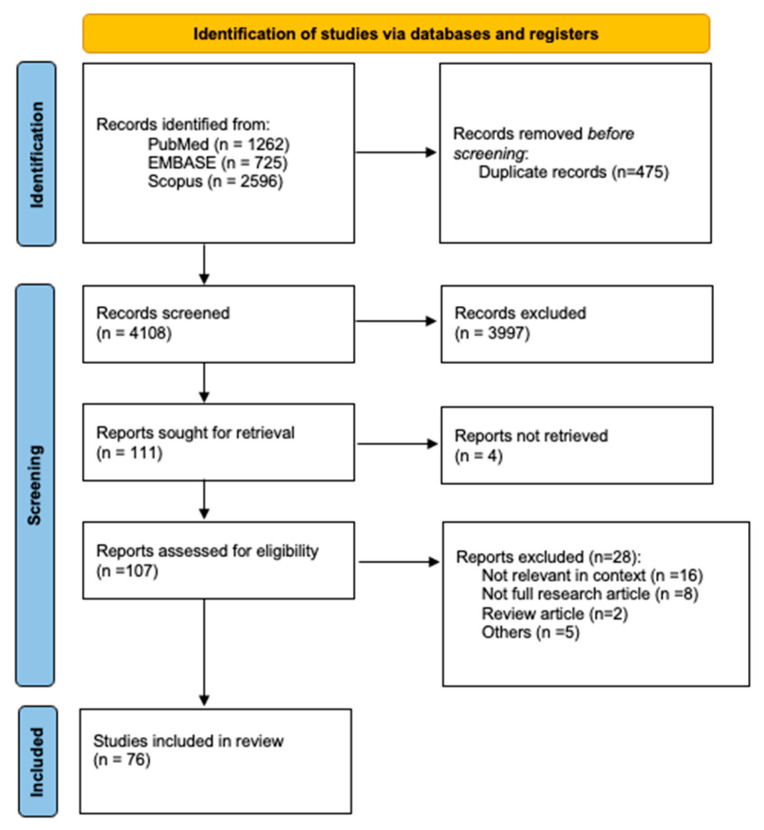
PRISMA flow diagram.

**Figure 2 ijms-25-01258-f002:**
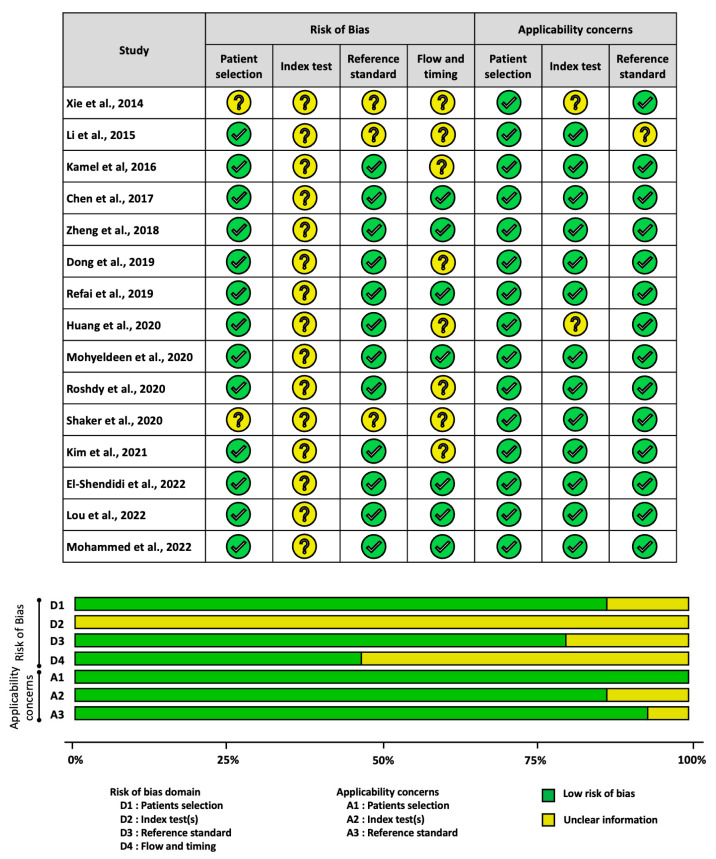
Quality assessment in articles investigating candidate lncRNAs [24,25,26,27,28,29,30,31,32,33,35,36,37,38].

**Figure 3 ijms-25-01258-f003:**
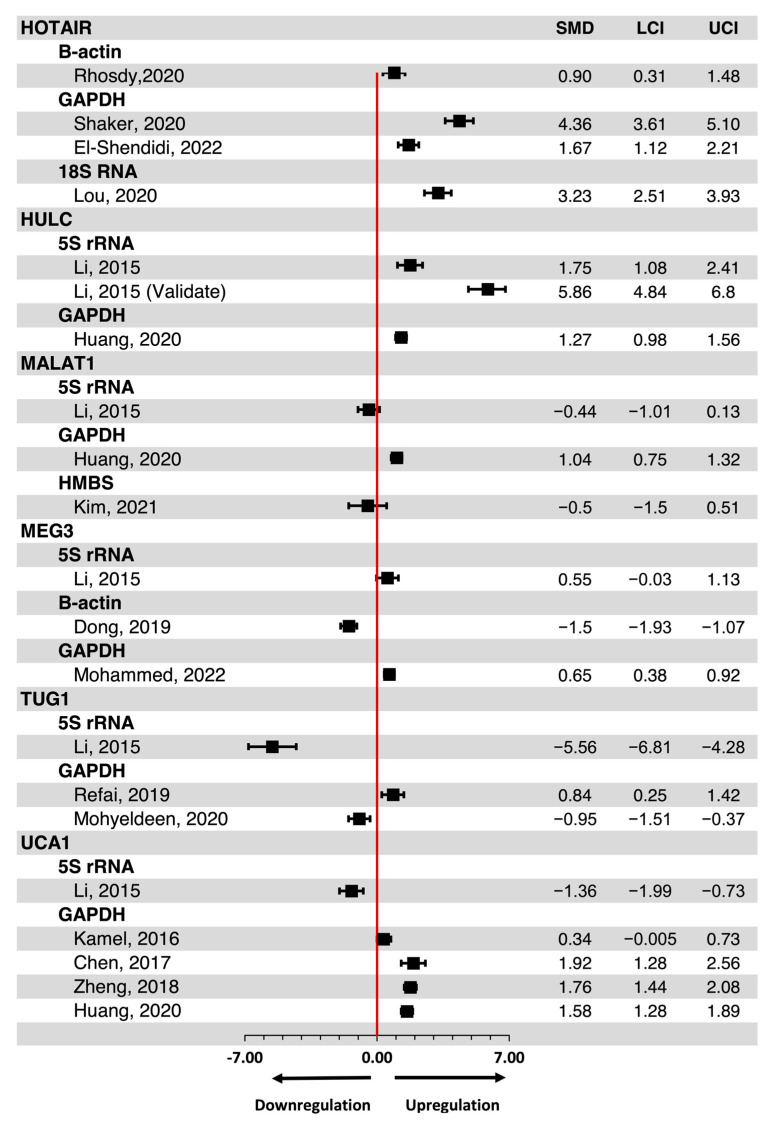
Expression level of candidate lncRNAs in HCC compared to healthy control. SMD—standardized mean difference; LCI—lower bound confidence interval; UCI—upper bound confidence interval. HOTAIR [24,25,26,27], HULC [29,30], MALAT1 [29,30,31], MEG3 [29,32,35], TUG1 [29,34,35], UCA1 [29,30,36,37,38].

**Figure 4 ijms-25-01258-f004:**
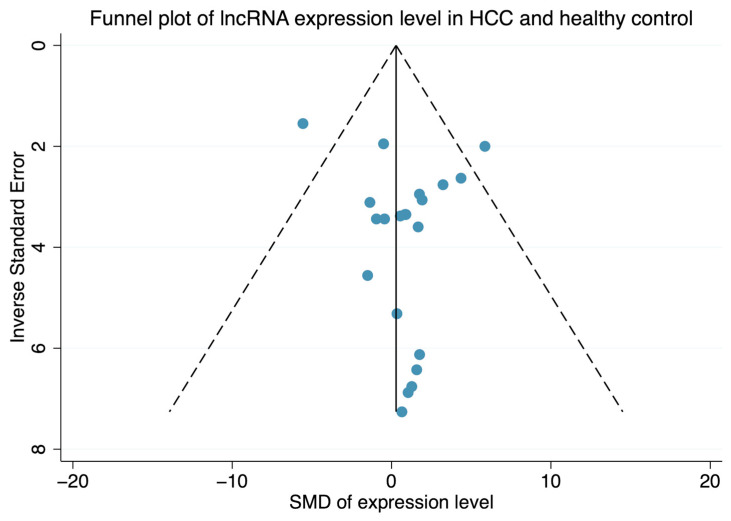
Funnel plot for assessment of publication bias.

**Figure 5 ijms-25-01258-f005:**
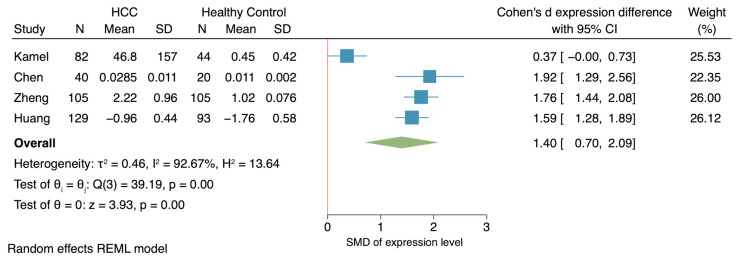
Meta-analysis of UCA1 expression level [30,36,37,38]. The standardized mean difference (SMD) was calculated from expression level of UCA1 in patients with HCC compared to healthy control. Blue boxes indicate point estimation and the whisker represented 95% confident interval. Green diamond represents pooled effect from four studies and red line refers to no difference in expression level.

**Figure 6 ijms-25-01258-f006:**
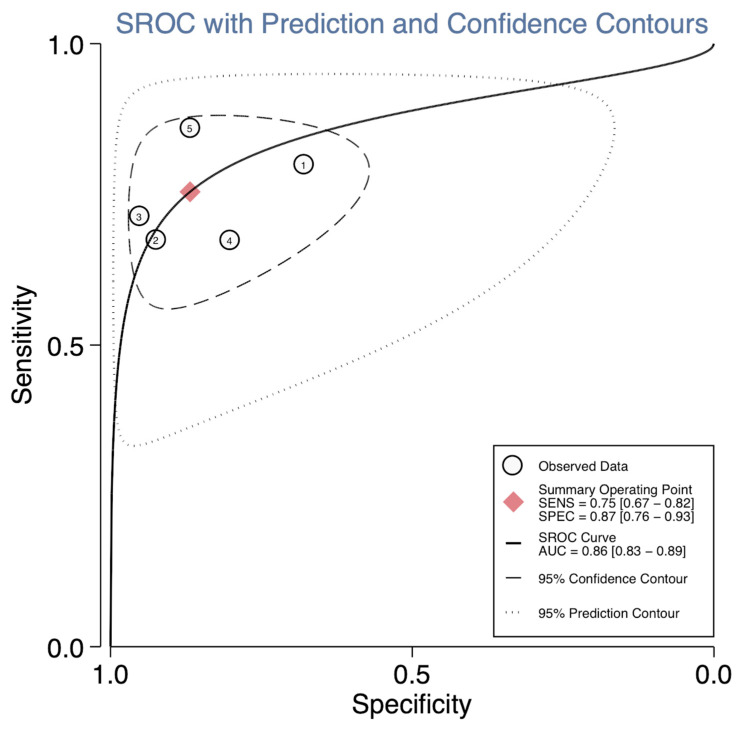
Summary receiver operating characteristic curve of candidate lncRNAs from HCC and liver disease patients. Studies 1–5 are labeled according to study no. in Table 2. Diagnostic OR, 20 (95% CI, 10–42). I^2^, 81% (95% CI, 60–100).

**Table 1 ijms-25-01258-t001:** Characteristics of studies with replicated investigation on candidate lncRNAs.

First Author, Year (Country)	House Keeping Gene	Case/Control Group	Sample Size (Case/Control)	auROC (95% CI)	Cut-Off Value	Sensitivity (%)	Specificity (%)
HOTAIR							
Rhosdy F, 2020 (Egypt) [24]	B-actin	HCC/LD	25/50	0.74	9.20	64.00	86.00
		HCC/cirr	25/25	0.70	7.00	64.00	76.00
		Cirr/non-Cirr	25/25	0.52	4.70	48.00	72.00
Shaker OG, 2020 (Egypt) [25]	GAPDH	HCC/HCV	50/50	0.78	3.13	80.00	68.00
El-Shendidi A., 2022 (Egypt) [26]	GAPDH	HCC stage AB/cirr	40/40	0.82 (0.73–0.89)	9.42	67.50	93.30
		HCC stage CD/HCC stage AB	40/40	0.71 (0.56–0.90)	15.45	66.00	78.00
Lou Z., 2022 (China) [27]	18S rna	HCC/HC	61/20	0.99 (0.98–1.00)	0.49 × 10^−4^	96.70	95.00
		HCC/cirr	61/20	0.81 (0.71–0.91)	1.45 × 10^−4^	59.00	100.00
HULC							
Xie, 2014 (China) [28]	GAPDH	HCC/HC	30/20	0.86	NA	NA	NA
Li, 2015 (China) [29]	5S rRNA	HCC/HC	24/24	NA	NA	NA	NA
Li, 2015 (China); validation [29]	5S rRNA	HCC/HC	66/24	0.78	NA	NA	NA
Hunag J, 2020 (China) [30]	GAPDH	HCC/HC	129/93	0.80 (0.73–0.86)	NA	86.00	62.40
		HCC/other	129/169	0.76 (0.70–0.81)	NA	86.00	55.60
MALAT1							
Li, 2015 (China) [29]	5S rRNA	HCC/HC	24/24	NA	NA	NA	NA
Huang J, 2020 (China) [30]	GAPDH	HCC/HC	129/93	0.77 (0.71–0.83)	NA	59.70	80.60
		HCC/other	129/169	0.73 (0.68–0.79)	NA	59.70	75.70
Kim SS, 2021 (Korea) [31]	HMBS	HCC/HC	7/9	NA	NA	NA	NA
MEG3							
Li, 2015 (China) [29]	5S rRNA	HCC/HC	24/24	NA	NA	NA	NA
Dong, 2019 (China) [32]	B-actin	HCC/HC	54/54	NA	NA	NA	NA
Mohammed S.R., 2022 (Egypt) [33]	GAPDH	HCC/HC	114/110	0.72 (0.64–0.81)	0.98	72.20	100.00
TUG1							
Li, 2015 (China) [29]	5S rRNA	HCC/HC	24/24	NA	NA	NA	NA
Refai NS, 2019 (Egypt) [34]	GAPDH	HCC/HC	30/20	NA	20.60	93.30	100.00
Mohyeldeen M, 2020 (Egypt) [35]	GAPDH	HCC/HC	40/20	0.96	40.00	90.00	92.30
		HCC/HCV	40/40	0.71	25.00	75.00	64.10
UCA1							
Li, 2015 (China) [29]	5S rRNA	HCC/HC	24/24	NA	NA	NA	NA
Kamel, 2016 (Egypt) [36]	GAPDH	HCC/HC	82/44	0.86 (0.80–0.92)	1.04	92.70	82.10
Chen, 2017 (China) [37]	GAPDH	HCC/HC	20/20	NA	NA	NA	NA
Zheng, 2018 (China) [38]	GAPDH	HCC/HC	105/105	0.90	1.85	73	99.00
		HCC/LD	105/105	0.85	1.99	71.4	94.30
Huang J, 2020 (China) [30]	GAPDH	HCC/HC	129/93	0.86 (0.81–0.91)	NA	81.40	75.30
		HCC/others	129/169	0.81 (0.76–0.86)	NA	67.40	80.50

Abbreviation: auROC—area under receiver operating characteristics curve; BLD—benign liver disease; cirr—cirrhosis; HCC—hepatocellular carcinoma; HC—healthy control; HCV—hepatitis C virus; IQR—interquartile range; LD—liver disease; med—median; NA—not available; SD—standard deviation.

**Table 2 ijms-25-01258-t002:** Diagnostic indices extracted from candidate lncRNAs for discrimination of HCC and liver disease patients.

Study No.	Author, Year	lncRNA	HCC Case (n)	LD (n)	Diagnostic Indices			
					TP	FN	TN	FP
1	Shaker, 2020 [25]	HOTAIR	50	50	40	10	34	16
2	El-Shendidi, 2022 [26]	HOTAIR	40	40	27	13	37	3
3	Zheng, 2018 [38]	UCA1	105	105	75	30	100	5
4	Huang, 2020 [30]	UCA1	129	76	87	42	61	15
5	Huang, 2020 [30]	HULC	129	76	111	18	66	10

Abbreviation: FN—false negative; FP—false positive; HCC—hepatocellular carcinoma; LD—liver diseases; TN—true negative; TP—true positive.

## Data Availability

All data used to support the findings of this study are available within this manuscript and the Appendix A.

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
