# Peer review of "Combinatorial Gene Expression Profiling of Serum HULC, HOTAIR, and UCA1 lncRNAs to Differentiate Hepatocellular Carcinoma from Liver Diseases: A Systematic Review and Meta-Analysis"

_ijms, 2024, doi:10.3390/ijms25021258_

Round 1
Reviewer 1 Report
Comments and Suggestions for Authors
Authors provided an interesting meta-analysis and systemic review on lncRNA for distinguishing between healthy controls, LDs and HCC. Study is well organized and results interesting. I have only a few minor comments:
1. Limited number of studies demomstrates potential bias on the results, such as Kamel or Li study. Risk should be better described.
2. Funnel plots should be presented, at lest the most important.
3. Table 1 is voluminious, please overthink how to compress it and make more readable.
Author Response
Reviewer #1:
Authors provided an interesting meta-analysis and systemic review on lncRNA for distinguishing between healthy controls, LDs and HCC. Study is well organized and results interesting. I have only a few minor comments:
- Limited number of studies demomstrates potential bias on the results, such as Kamel or Li study. Risk should be better described.
Response:
Thank you for pointing out this issue. Certainly, there was some error regarding the unmatched of data information in table S9 and in the original Figure 2. Accordingly, we have amended both the figure 2 and text. Please notice that the NEW figure 2 has changed in Li et al., 2015; Kamel et al., 2016; Chen et al., 2017 Dong et al., 2019; and Huang et al., 2020 studies.
Also, we have elaborated more on the potential bias in results in the 3.2 Quality of Evidence among Candidate lncRNAs section in lines 203-223 as follows :
“The quality of eligible studies was determined based on QUADAS-2 across 4 risk of bias domains and 3 applicability concerns (Figure 2). All studies adopted a case-control design, which is generally necessary for primary investigations into lncRNA expression levels. However, the interpretation of results raised some concerns. Since the diagnostic study required measurements in all consecutive patients suspected of having the disease to prevent potential bias, the use of a case-control design may result in an overestimation of the outcomes [23].
Additionally, concerning the index test, none of the studies pre-specified the appropriate cut-off (threshold) point for diagnostic accuracy analyses. Consequently, the results exhibited high heterogeneity, and were considered data-driven analyses by the nature of study.
Furthermore, a few studies (13.3%) did not provide adequate diagnostic criteria for patients with HCC and liver disease. Most healthy subjects did not undergo similar reference tests, primarily because these tests are invasive, such as liver biopsy. Additionally, one study (Kim SS et al., 2021) excluded certain patients from the final analysis. Notably, 7 studies (46.7%) (Xie et al., 2014; Li et al., 2015; Kamel et al., 2016; Dong et al., 2019; Huang et al., 2020; Shaker et al., 2020; Kim et al., 2021) did not clearly specify whether the HCC samples were collected before any treatment.
In terms of applicability, more than 80% of studies provided evidence that matches our research question across 3 applicability domains (Supplementary table S9).”
- Funnel plots should be presented, at least the most important.
Response:
We agree with the Reviewer that funnel plots should be presented. Accordingly, we have generated the funnel plot for publication bias of included studies with candidate lncRNAs and presented it as NEW Figure 4 in the manuscript. Moreover, we have added the Egger’s test of small study effects in the section 3.3 pag 8 lines 252-253 of the revised manuscript, as follow:
“Of these studies, no significant publication bias observed when Egger’s test of small study effects was performed (Figure 4; p-value 0.208).”
- Table 1 is voluminious, please overthink how to compress it and make more readable.
Response:
We appreciate the feedback from the reviewer regarding the original Table 1, which was extensive. In response, we have restructured Table 1 for improved readability. Furthermore, to facilitate access to all the original details from the extensive Table 1, we have introduced a new supplementary Table S8.

Reviewer 2 Report
Comments and Suggestions for Authors
The manuscript sought to identify and analyse the relevant diagnostic characteristics of circulating lncRNAs in hepatocellular carcinoma. The subject matter is current and will captivate the readership of the journal.Nevertheless, the paper is deficient in both data and a compelling theoretical framework, prerequisites for publication consideration.
The research primarily sought to answer the question of whether hepatocellular carcinoma is caused by a persistent infection with the hepatitis B or C virus.Some factors that can speed up the course of this disease include cirrhosis, excessive alcohol consumption, aflatoxin-contaminated foods, non-alcoholic fatty liver disease, smoking, and non-alcoholic fatty liver disease. In multiple aspects, the authors should be included in the methods section. However, it is crucial to conduct a practical trial, either in a laboratory or in a study field, and then establish a connection with previous research. ​Because increased serum levels of HULC, HOTAIR, and UCA1 in HCC patients are not the sole indicator, the results were in line with the offered evidence and arguments but failed to answer the primary question.
In my opinion, the authors should employ a more strategic approach in determining the claims they can assert, taking into consideration the data at their disposal. This will enable them to transform their study into a valuable academic contribution. Regrettably, based on its current condition, I cannot endorse the acceptance of this paper.
Author Response
Reviewer #2 :
The manuscript sought to identify and analyse the relevant diagnostic characteristics of circulating lncRNAs in hepatocellular carcinoma. The subject matter is current and will captivate the readership of the journal.Nevertheless, the paper is deficient in both data and a compelling theoretical framework, prerequisites for publication consideration.
The research primarily sought to answer the question of whether hepatocellular carcinoma is caused by a persistent infection with the hepatitis B or C virus.Some factors that can speed up the course of this disease include cirrhosis, excessive alcohol consumption, aflatoxin-contaminated foods, non-alcoholic fatty liver disease, smoking, and non-alcoholic fatty liver disease. In multiple aspects, the authors should be included in the methods section. However, it is crucial to conduct a practical trial, either in a laboratory or in a study field, and then establish a connection with previous research. ​Because increased serum levels of HULC, HOTAIR, and UCA1 in HCC patients are not the sole indicator, the results were in line with the offered evidence and arguments but failed to answer the primary question.
In my opinion, the authors should employ a more strategic approach in determining the claims they can assert, taking into consideration the data at their disposal. This will enable them to transform their study into a valuable academic contribution. Regrettably, based on its current condition, I cannot endorse the acceptance of this paper.
Response :
We thank the Reviewer for thoroughly evaluate our manuscript.
It seems that this Reviewer is posing two major concerns: 1) The manuscript is criticized for lacking both sufficient data and a compelling theoretical framework. 2) The increased serum levels of HULC, HOTAIR, and UCA1 in HCC patients, while discussed, are criticized for “not being the sole indicator and for failing to conclusively answer the primary question”.
As for the point 1, we have previously outlined the scope of our research in the introduction section on page 2, lines 79-89. In summary, we sought to determine whether, given the presently available public information on serum lncRNAs in hepatocellular carcinoma, it is feasible to identify non-invasive biomarkers for implementation as diagnostic tools in clinical practice. Data were collected using a systematic and unbiased approach, which has been registered with the International Prospective Register of Systematic Reviews (PROSPERO: CRD42022363196) and is presented step by step in the methods section.This includes for instance: Systematic Searching and Eligible Criteria (pag 3 line 99), Outcome of Interest (pag 2 line 114), Data Extraction (page 3 line 124), Quality Assessment (pag 3 line 135).
The scope of this study was further elucidated in the discussion on page 12, line 345, where we affirmed: "Our study introduces an updated panel of circulating lncRNAs that shows promise for future integration into clinical practice for the early diagnosis of HCC."
As such, it is evident that this study does not present any clinical validation of the proposed serum lncRNAs performed by our team. As we have also mentioned at page 12 lines 352-354 (Therefore, to establish the comprehensive diagnostic utility of lncRNAs in HCC, it is imperative to await further relevant studies and engage in more extensive data analysis).
Finally, our effort led to the identification of three serum lncRNAs whose expression have been statistically analyzed in comparison to the biomarkers currently used in clinical practice for HCC diagnosis.
We acknowledge that drawing comprehensive overarching conclusions from this array of diverse measurements is not practically feasible. Nevertheless, these outcomes do indicate an encouraging trend toward the utilization of lncRNAs as potential diagnostic biomarkers, as we mentioned at pag 13 lines 365-369.
Moreover, we offered our opinion on the possible implication of this study at page 13 lines 375-403.

Round 2
Reviewer 2 Report
Comments and Suggestions for Authors
The manuscript has been revised and is now being accepted for publication in our journal. Significant gaps in the manuscript have been corrected by the author. I thus accepted publication.